# Thalamic input to auditory cortex is locally heterogeneous but globally tonotopic

**Sebastian A Vasquez-Lopez[1], Yves Weissenberger[1], Michael Lohse[1], Peter Keating[1,2], Andrew J King[1], Johannes C Dahmen[1]\***

[1]Department of Physiology, Anatomy and Genetics, University of Oxford, Oxford, United Kingdom; [2]Ear Institute, University College London, London, United Kingdom

**Abstract** Topographic representation of the receptor surface is a fundamental feature of sensory cortical organization. This is imparted by the thalamus, which relays information from the periphery to the cortex. To better understand the rules governing thalamocortical connectivity and the origin of cortical maps, we used in vivo two-photon calcium imaging to characterize the properties of thalamic axons innervating different layers of mouse auditory cortex. Although tonotopically organized at a global level, we found that the frequency selectivity of individual thalamocortical axons is surprisingly heterogeneous, even in layers 3b/4 of the primary cortical areas, where the thalamic input is dominated by the lemniscal projection. We also show that thalamocortical input to layer 1 includes collaterals from axons innervating layers 3b/4 and is largely in register with the main input targeting those layers. Such locally varied thalamocortical projections may be useful in enabling rapid contextual modulation of cortical frequency representations.

DOI: https://doi.org/10.7554/eLife.25141.001

## Introduction

The vast majority of ascending sensory information reaches the cortex via the thalamus. To understand the functional organization of cortical circuits, it is therefore crucial to uncover the rules of thalamocortical connectivity (*Jones, 2007*; *Sherman and Guillery, 2013*; *Winer et al., 2005*). Most of the brain's auditory neurons, including those in the medial geniculate body (MGB) of the thalamus, are tuned to sound frequency and their spatial arrangement reflects the tonotopic organization established by the biomechanical properties of the cochlea. Tonotopy is preserved across species and at every lemniscal stage of the ascending auditory pathway up to the cortex (*Kaas, 2011*; *Schreiner and Winer, 2007*; *Woolsey and Walzl, 1942*). Like the receptor surface maps that are also hallmarks of the visual and somatosensory pathways, the presence of sound frequency gradients within each of these brain regions is therefore the most well characterized feature of the auditory system.

While the existence of cortical tonotopy is universally accepted, how precise this organization really is has recently been debated (*Guo et al., 2012*; *Kanold et al., 2014*; *Rothschild and Mizrahi, 2015*). In particular, the opportunity to image the activity of large populations of neurons at single-cell resolution in the mouse auditory cortex (*Bandyopadhyay et al., 2010*; *Issa et al., 2014*; *Rothschild et al., 2010*; *Winkowski and Kanold, 2013*) has questioned the smooth tonotopic organization revealed with microelectrode recordings (*Guo et al., 2012*; *Hackett, 2011*; *Stiebler et al., 1997*) or low-resolution imaging methods (*Horie et al., 2013*; *Moczulska et al., 2013*; *Tsukano et al., 2016*). The current view holds that neurons in the main thalamorecipient layers 4 and 3b, which tend to be most commonly sampled by microelectrode recordings, exhibit precise

**\*For correspondence:**
johannes.dahmen@dpag.ox.ac.uk

tonotopy that transitions into a coarse and more heterogeneous frequency organization in the supra-granular layers (*Kanold et al., 2014*).

One implication of this arrangement is that the homogenous tonotopy of the middle cortical layers is inherited from thalamic input which is itself precisely tonotopically ordered. However, it is unclear how tightly organized this projection actually is. Although retrograde tracing of thalamocortical inputs (*Brandner and Redies, 1990*; *Hackett et al., 2011*) suggests strict topography, anterograde tracing (*Huang and Winer, 2000*) and reconstruction of single thalamic axons (*Cetas et al., 1999*) indicate considerable divergence in the auditory thalamocortical pathway. Indeed the frequency tuning of thalamic inputs that converge onto individual auditory cortical neurons can span several octaves (*Liu et al., 2007*), suggesting a need for integration across differently tuned afferent terminals. Furthermore, while most thalamocortical projections target the middle cortical layers, axons from the MGB can also be found in other layers (*Frost and Caviness, 1980*; *Huang and Winer, 2000*; *Ji et al., 2016*; *Kimura et al., 2003*; *Llano and Sherman, 2008*), but nothing is currently known about the relative specificity or precision of these inputs.

Our current understanding of the functional organization of the auditory thalamocortical pathway is limited by the relatively poor spatial resolution of the methods that have so far been used to investigate it. In this study, we employed in vivo two-photon (*Denk et al., 1990*) axonal calcium imaging (*Glickfeld et al., 2013*; *Petreanu et al., 2012*; *Roth et al., 2016*) to measure for the first time the frequency selectivity of individual boutons on auditory thalamocortical axons. Across different anesthetic states and different strains of mice, we found that the tuning of neighboring boutons is surprisingly heterogeneous, that frequency gradients are apparent at a large spatial scale only, and that thalamic inputs to cortical layers 1 and 3b/4 share a similarly coarse tonotopic organization. Furthermore, we demonstrate that this organization, which provides a potential basis for the broad spectral integration and experience-dependent plasticity that are characteristic features of the tuning properties of auditory cortical neurons, reflects almost exclusively the properties of the lemniscal thalamocortical projection originating in the ventral division of the medial geniculate body.

## Results

We initially expressed GCaMP6m (*Chen et al., 2013*) throughout the auditory thalamus (*Figure 1A*) in order to functionally characterize its input to the auditory cortex. Most of the thalamic axons were found in the middle layers, L3b/4, but substantial input was also observed in L1. We measured calcium transients, which correlate with somatic spiking activity (*Petreanu et al., 2012*), in individual putative synaptic boutons of thalamocortical axons (*Figure 1B*) in L1 and L3b/4 of anesthetized mice during presentation of pure tones and assessed their frequency sensitivity (*Figure 1C*).

For each 90 × 100 µm region of auditory cortex we recorded from dozens to hundreds of tone-responsive (L1: 90.5 ± 64 (median ± interquartile range), n = 36 imaged regions; L3b/4: 87 ± 84, n = 36) and mostly well-tuned boutons (*Figure 1—figure supplement 1*). Given that the auditory cortex is tonotopically organized and that this organization must be inherited from the thalamus, the cortex's sole source of ascending auditory information, we expected the thalamic input to be tightly tonotopically ordered. Consequently, when sampling from a small patch of cortex, the boutons found therein ought to be tuned to similar frequencies. To our surprise, even neighboring boutons could be tuned to frequencies several octaves apart both in L1 and L3b/4 (*Figure 1D–G*). In order to quantify the variation in frequency selectivity among a population of nearby thalamocortical boutons, we determined each bouton's best frequency (BF), defined as the frequency at which the strongest response occurred in the level-averaged tuning curve (*Guo et al., 2012*) (*Figure 1F–I*), measured the co-tuning (the standard deviation of the BF distribution) for each imaged region (*Figure 1J*) and compared regions recorded at depths corresponding to L1 (55 ± 39 µm) with those recorded at the same x-y coordinates but at depths corresponding to L3b/4 (311 ± 43.5 µm) (*Figure 1K*). While the average co-tuning of thalamic boutons in a 90 × 100 µm region of auditory cortex was about one octave, there was a slight, but statistically significant, difference between the inputs to the different layers. Input to L3b/4 shows stronger co-tuning (0.93 ± 0.25 octaves) and is, thus, more homogeneous than the input to L1 (1.15 ± 0.37 octaves, p<0.001, effect size: r = 0.43, n = 36, Wilcoxon signed-rank test, *Figure 1J*). Within L3b/4 there was no relationship between depth and co-tuning (R = 0.14, p=0.41, n = 36, Spearman's correlation), which suggests that layer 4 is no more homogeneous than lower layer 3 (*Figure 1—figure supplement 2*).

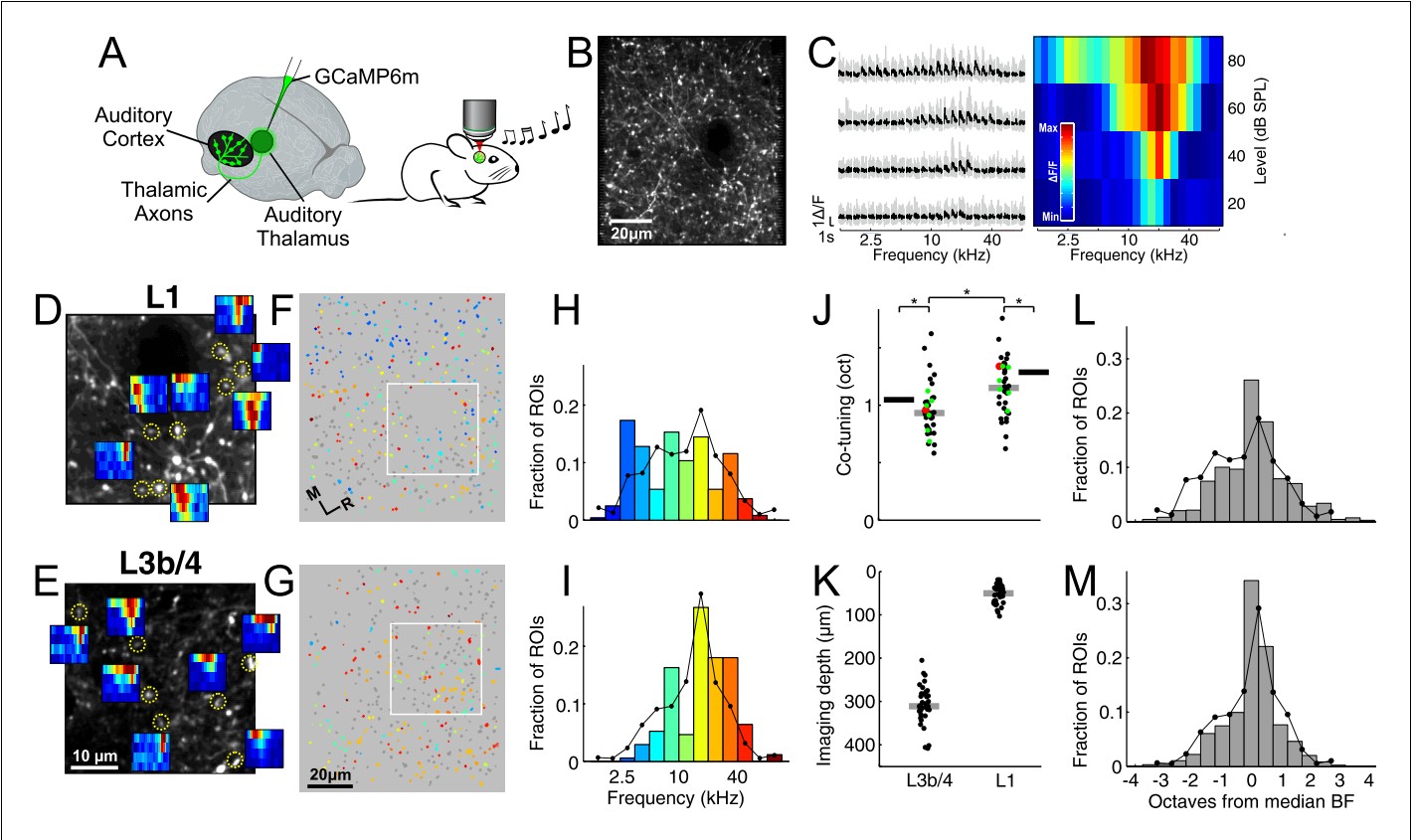

**Figure 1.** Local heterogeneity of thalamic input. (**A**) Experimental schematic. Responses of thalamocortical axons in auditory cortex to pure tones were imaged using two-photon microscopy in anesthetized mice expressing the calcium indicator GCaMP6m in the auditory thalamus. (**B**) In vivo two-photon image of thalamic axons and axonal boutons in auditory cortex. (**C**) Left, example fluorescence traces of one bouton in response to randomized pure tones, shown here ordered according to sound frequency and level. Gray traces indicate responses to individual repetitions. Black traces indicate mean responses. Right, frequency response area corresponding to traces on left. Level-averaged tuning curves could generally be well approximated by a Gaussian (**Figure 1—figure supplement 1**). (**D**) In vivo two-photon image of thalamic axons and boutons in a small patch of L1 of the auditory cortex. FRAs are shown for several example boutons (locations indicated by yellow circles). (**E**) Same as **D** for L3b/4. (**F**) ROIs corresponding to putative L1 thalamocortical boutons from a single optical plane color-coded according to each bouton's BF. Non-responsive ROIs are shown in dark gray. White rectangle corresponds to area shown in (**D**). (**G**) Same as **F** for region imaged in L3b/4. (**H**) Distribution of BFs from the L1 region shown in **F**. Line shows overall BF distribution of all L1 boutons pooled from all imaged regions and animals. (**I**) Same as **H** for L3b/4. (**J**) Co-tuning (standard deviation of BF distribution) within individual regions imaged in L1 (n = 36) and L3b/4 (n = 36). Thick gray lines indicate medians. Thick black lines indicate co-tuning for overall BF distributions (black lines in **H,I**). Red dots indicate co-tuning of regions shown in **F** and **G**. Green dots indicate co-tuning of regions shown in **Figure 2C,D**. (**K**) Depth of all regions imaged in L1 and L3b/4. Thick gray lines indicate medians. There was no relationship between imaging depth within L3b/4 and co-tuning (**Figure 1—figure supplement 2**). (**L**) Bar graph shows the average of the normalized BF distributions for L1. In order to produce this average distribution the BF distributions of individual regions (such as the one in **H**) were normalized by setting the median BF to zero before averaging. Line re-plots the overall BF distribution of all pooled L1 boutons shown in **H**. (**M**) Same as **L** for L3b/4.

DOI: https://doi.org/10.7554/eLife.25141.002

The following figure supplements are available for figure 1:

**Figure supplement 1.** Frequency tuning of thalamocortical boutons.
DOI: https://doi.org/10.7554/eLife.25141.003

**Figure supplement 2.** No relationship between imaging depth within L3b/4 and co-tuning.
DOI: https://doi.org/10.7554/eLife.25141.004

Given the size of the imaged regions relative to the size of the auditory cortex and its subfields, the variation in frequency tuning appeared unexpectedly large so, for comparison, we pooled all boutons across all imaged regions and animals to obtain overall BF distributions for L1 and L3b/4. Mice are sensitive to frequencies between about 1 and 100 kHz (**Willott, 2001**), but their brains do not represent all frequencies within that range equally. Most auditory nerve fibers (**Ehret, 1979**) and most neurons in the inferior colliculus (**Stiebler and Ehret, 1985**) and thalamus (**Anderson and**

*Linden, 2011*) are tuned to frequencies in the middle one to two octaves of the mouse's hearing range. Consistent with the frequency distributions reported in the inferior colliculus and thalamus, we found that the overall BF distribution of thalamocortical inputs had a pronounced bias towards frequencies near the center of the mouse's hearing range (*Figure 1H,I*).

If the thalamic input exhibits a tight tonotopic organization, the BF distributions of individual imaged regions should be much more narrow, that is they should show stronger co-tuning, than the overall BF distribution. If there is no relationship between spatial position and frequency, the BF distributions of individual regions (*Figure 1L,M*, bars) should resemble the overall BF distribution (*Figure 1L,M*, lines). We found that the overall BF distributions for L1 (1.29 octaves) and L3b/4 (1.05 octaves) exhibited slightly but significantly weaker co-tuning than individual imaged regions (L1: p=0.003, effect size: r = 0.50, n = 36; L3b/4: p=0.014, effect size: r = 0.42, n = 36, Wilcoxon signed-rank test), indicating some selectivity in the BFs represented within imaged regions. Furthermore, the difference in co-tuning between pairs of individual L1 and L3b/4 regions (*Figure 1J*) could be accounted for by the difference between the overall BF distributions for L1 and L3b/4 (co-tuning of boutons within imaged regions / co-tuning of overall BF distribution: for L1 = 89.4 ± 29.1%; for L3b/4 = 88.9 ± 24.3%, p=0.56, n = 36, Wilcoxon signed-rank test).

If the difference in co-tuning between individual regions and the overall BF distribution is the result of tonotopic organization, then neighboring boutons —even within a small patch of cortex— should be more similar in their tuning than topographically distant ones. Indeed, we observed a relationship between topographic distance and the difference in BF (*Figure 2A*). Interestingly, this very small but statistically significant correlation was present not only in the main thalamic input to L3b/4 (R = 0.034, p<10$^{-45}$, for all possible pairs of boutons, n = 167521, Spearman's correlation), but also in L1 (R = 0.035, p<10$^{-63}$, n = 237107, Spearman's correlation), suggesting that input to L1 has a similar degree of topographic order. The relationship between distance and frequency selectivity was not simply the result of a topographic clustering of boutons from the same axon because the correlation remained even when pairs with the same BF were excluded from the analysis (L3b/4: R = 0.027, p<10$^{-24}$, n = 142108; L1: R = 0.024, p<10$^{-28}$, n = 210001, Spearman's correlation).

Next we asked whether the inputs to L1 and L3b/4 are in register. We found that there is a close correspondence between the mean BF of a region imaged in L3b/4 and the mean BF of one imaged in L1 immediately above, suggesting that the two input channels are matched tonotopically (*Figure 2B*, R = 0.59, p=0.0002, n = 36, Pearson correlation). Finally, we examined whether, on a more global scale spanning several hundred micrometers of cortex and several imaged regions, tonotopic gradients might become apparent. *Figure 2C* illustrates the results from an experiment in which gaps in the vasculature allowed us to image several regions close together. The caudo-rostral low-to-high tonotopic gradient indicative of mouse A1 now emerged both in the inputs to L3b/4 and the inputs to L1. Furthermore, the co-tuning in these regions (green dots in *Figure 1J*) was representative of the co-tuning of the entire sample, suggesting either that most of the data were collected in A1 or that the co-tuning of the thalamic input is similar across cortical fields.

The C57BL/6 strain employed in the above experiments is the most popular laboratory mouse strain, and is used as genetic background for the overwhelming majority of genetically modified mouse strains, the availability of which make this species such a useful model system for neuroscience research. C57BL/6 mice are not normally considered to suffer from impaired hearing at the age used here (*Ison et al., 2007*), but there have been some reports that a decline in the number of neurons tuned to high frequencies can be detected as early as 1–2 months after birth, especially at higher levels of the auditory pathway such as the cortex (*Willott et al., 1993*). We therefore carried out additional experiments on a novel C57BL/6 strain in which the Cdh23$^{ahl}$ allele that otherwise predisposes this strain to age-related high frequency hearing loss has been corrected (*Mianné et al., 2016*). Furthermore, and in order to rule out that any of the above reported results are dependent on the effects of anesthesia, we carried out these experiments in awake, passively listening animals.

While the C57BL/6NTac.Cdh23$^{753A>G}$ mice also showed a bias for frequencies near the middle of their hearing range, the proportion of high frequency BFs was greater than in the C57BL/6 mice and the overall BF distribution, thus, broader (*Figure 3A,B*). Overall, the median number of tone-responsive boutons obtained per imaging region was lower (29.5 ± 23), which could be partly due to the effects of anesthesia vs wakefulness. However, this might also reflect strain differences or other differences in methodology, such as the fact that we tended to image these animals slightly sooner after the virus injections (3–4 weeks), but over several days rather than in a single session

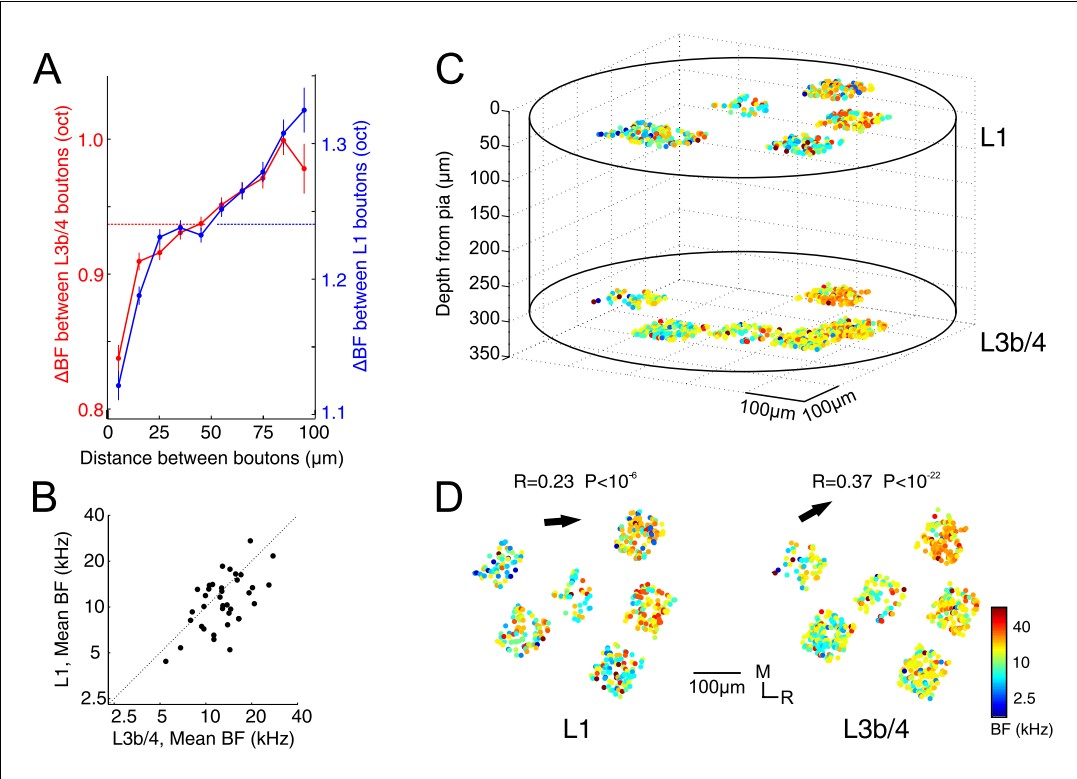

**Figure 2.** Tonotopic organization of thalamic input to auditory cortex. (**A**) Pairwise difference in BF (ΔBF) between boutons as a function of topographic distance in L1 (blue) and L3b/4 (red) for all possible bouton pairs. Number of pairs per 10 μm wide bin is between 2330 and 36725. Horizontal lines indicate average ΔBF across all bouton pairs for L1 (blue) and L3b/4 (red). (**B**) Mean BF of individual imaged regions in L1 versus mean BF of regions in L3b/4. (**C**) Relative spatial locations of tone-responsive boutons from several regions in L1 and L3b/4 of the same animal reconstructed in 3D space and color-coded according to each bouton's BF. (**D**) Top view of the boutons shown in C separated into L1 (left) and L3b/4 (right). Arrows indicate direction of tonotopic axis in L1 and L3b/4.
DOI: https://doi.org/10.7554/eLife.25141.005

immediately after the window implantation. Otherwise, the results were remarkably similar. Thus, the average co-tuning per imaged region of auditory cortex was just above one octave both near the cortical surface and in the middle layers (*Figure 3C*) with slightly stronger co-tuning in L3b/4 (1.21 ± 0.64 octaves) than in L1 (1.42 ± 0.50 octaves, p=0.011, effect size: r = 0.40, n = 20, Wilcoxon signed-rank test, *Figure 3D*). Moreover, as in the preceding experiments, the inputs to L1 and L3b/4 were matched tonotopically (*Figure 3E*, R = 0.67, p=0.0013, n = 20, Pearson correlation). Where it was possible to image several regions over a large enough area, the caudo-rostral, low-to-high and high-to-low tonotopic gradients that are respectively indicative of A1 and the anterior auditory field (AAF), the primary cortical areas of the mouse, emerged (*Figure 3F*). We followed up these experiments with microelectrode recordings to obtain cortical multi-unit frequency maps that helped us to attribute individual imaging regions to particular cortical fields even in those cases when the thalamic input frequency maps were inconclusive (*Figure 3G*). These recordings demonstrated that the vast majority (18/20) of imaging regions were located in the primary cortical areas.

The auditory thalamus consists of several subnuclei. Besides the ventral division of the MGB (MGBv), which is the largest subnucleus and part of the lemniscal pathway, these are the non-lemniscal dorsal division of the MGB (MGBd) and the paralaminar nuclei —the medial division of the MGB (MGBm), the posterior intralaminar nucleus (PIN), the suprageniculate nucleus (SG) and peripeduncular nucleus (PP). Our imaging experiments were designed to characterize the full extent of the auditory thalamic input available to auditory cortex. To better understand the contributions of the lemniscal, non-lemniscal and paralaminar subnuclei to the thalamocortical projection we next carried out a number of mostly anatomical experiments.

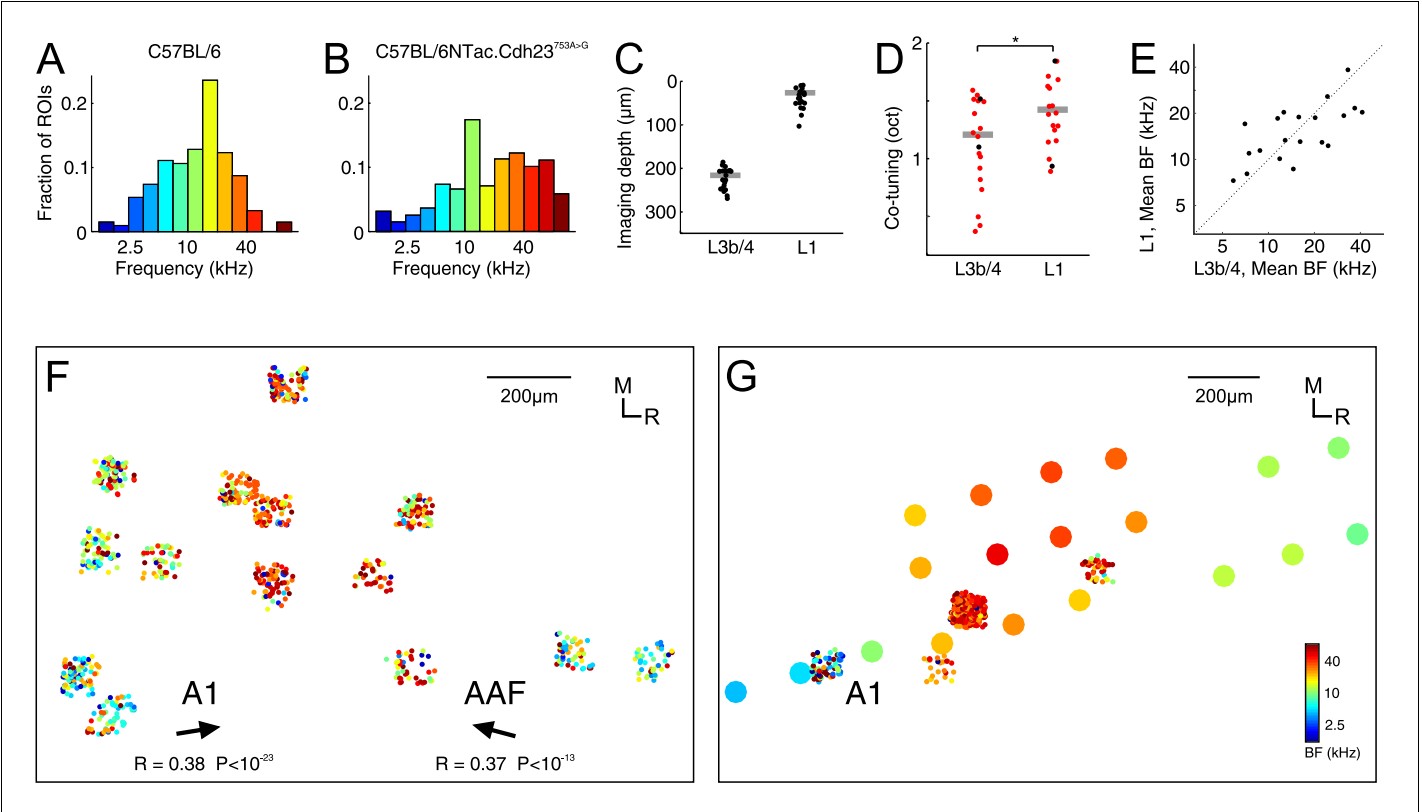

**Figure 3.** Characterization of thalamic input to the auditory cortex of awake C57BL/6NTac. $Cdh23^{753A>G}$ mice. (A) Overall BF distribution of all boutons from C57BL/6 mice. (B) Overall BF distribution of all boutons from C57BL/6NTac.$Cdh23^{753A>G}$ mice. (C) Depth of all imaged regions in C57BL/6NTac.$Cdh23^{753A>G}$ mice. (D) Co-tuning (standard deviation of BF distribution) within individual regions imaged in L1 (n = 20) and L3b/4 (n = 20) in C57BL/6NTac.$Cdh23^{753A>G}$ mice. Red dots indicate co-tuning of regions identified to be in a primary auditory field. (E) Mean BF of individual imaged regions in L1 versus mean BF of regions in L3b/4 in C57BL/6NTac.$Cdh23^{753A>G}$ mice. (F) Locations of thalamic boutons from one animal color-coded by BF and collapsed onto the same horizontal plane. Arrows indicate direction of tonotopic axis of boutons deemed to be in A1 (left) and AAF (right). (G) Locations of thalamic boutons (small dots) and multi-unit recordings (large dots) from another animal color-coded by BF and collapsed onto the same horizontal plane. The color of the large dots indicates the mean BF of all tone-responsive multi-units recorded under anesthesia with multi-electrode arrays at the same site following completion of awake imaging.

DOI: https://doi.org/10.7554/eLife.25141.006

In the mouse, calretinin (CR) has been identified as a useful marker for distinguishing between different parts of the thalamus. Among the auditory subnuclei, only neurons in the MGBd, MGBm, SG, PIN and PP, but not the MGBv, contain CR (*Lu et al., 2009*), so we injected a mixture of viruses driving the cre-dependent expression of a red fluorescent protein and the non-cre dependent expression of a green fluorescent protein into the auditory thalamus of a CR-IRES-cre (*Taniguchi et al., 2011*) mouse, which expresses cre recombinase only in CR+ neurons. While green labelled neurons were found throughout the auditory thalamus, red labelled neurons were found exclusively outside of the MGBv (*Figure 4A,B*), which confirmed that the CR-IRES-cre line is suitable for targeting the non-lemniscal and paralaminar nuclei of the auditory thalamus. We found that input to the cortex from the neurons in these nuclei is restricted mostly to secondary auditory areas, particularly the ventrally located area A2. The few axons found in primary auditory cortical areas were restricted mostly to layer 1 and the even fewer axons found in the middle layers were located primarily below the main thalamic input (*Figure 4C*). Projections to regions outside the auditory cortex were found mostly in the amygdala and striatum.

To better resolve the organization of thalamic axons in the auditory cortex we performed minute injections of a mixture of highly diluted cre-expressing and cre-dependent eGFP-expressing viruses in different parts of the auditory thalamus of C57BL/6 mice. Using this approach (*Chen et al., 2013*; *Xu et al., 2012*), we were able to transfect very small numbers of neurons (12-53) and could reveal

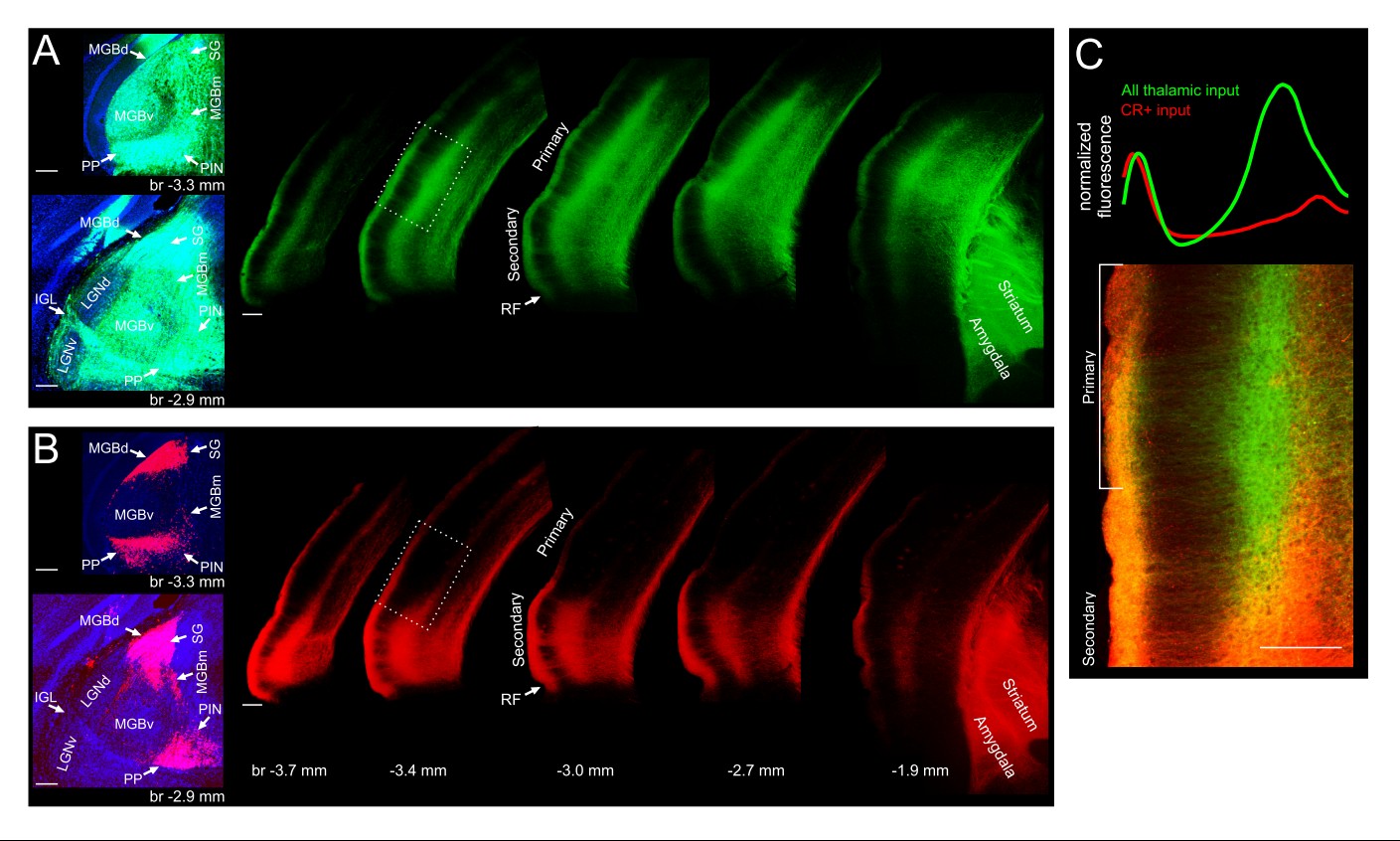

**Figure 4.** Input from non-lemniscal auditory thalamus targets almost exclusively non-primary auditory cortex. (A) Left, coronal sections showing GCaMP6m labelled neurons throughout the auditory thalamus of a CR-IRES-cre mouse injected with AAV1.Syn.GCaMP6m.WPRE.SV40 and AAV1.CAG. Flex.tdTomato.WPRE.bGH. Right, coronal sections showing GCaMP6m labelled thalamic axons in auditory cortex, amygdala and striatum. (B) Left, same coronal sections as in left panels of A, showing tdtomato labelled neurons exclusively in non-lemniscal and paralaminar nuclei. Right, same coronal sections as in right panel of A, showing tdtomato labelled thalamic axons almost exclusively in secondary auditory cortex, amygdala and striatum. (C) Bottom, overlay of GCaMP6m (all thalamic input) and tdtomato (CR+ input) labelled axons in transition area from secondary to primary auditory cortex indicated by white rectangle in A and B. Top, cortical depth profile of labelling from non-lemniscal thalamic axons (red, CR+) versus all thalamic axons (green) within primary auditory cortex (average across area within white bracket in bottom panel). The red and green lines were normalized to have same peak height in layer I. MGBd, dorsal division of medial geniculate body; MGBm, medial division of medial geniculate body; MGBv, ventral division of medial geniculate body; SG, suprageniculate nucleus; PIN, posterior intralaminar nucleus; PP, peripeduncular nucleus; LGNd, dorsal division of lateral geniculate nucleus; LGNv, ventral division of lateral geniculate nucleus; IGL, intergeniculate leaf; br, bregma; CR+, calretinin-positive; RF, rhinal fissure. Scale bars, 200 μm. Locations of thalamic subdivisions adopted from *Lu et al., 2009*. CR+ boutons typically responded very poorly to acoustic stimulation. See *Figure 4—figure supplement 1* for In vivo two-photon image of CR+ thalamic axons of CR-IRES-cre mouse injected with AAV1.Syn.Flex.GCaMP6m.WPRE.SV40 and example FRAs.

DOI: https://doi.org/10.7554/eLife.25141.007

The following figure supplement is available for figure 4:

**Figure supplement 1.** Calcium imaging of CR+ thalamic axons.

DOI: https://doi.org/10.7554/eLife.25141.008

that projections from the medial part of the auditory thalamus (MGBm/PIN) provide only extremely sparse input to auditory cortex (*Figure 5A*). This input primarily terminates in L1, and otherwise is located below the middle layer(s) where input from the MGBv is densest. Projections from the PP do not enter the auditory cortex and instead remain subcortical where they target amygdala, striatum and midbrain (*Figure 5B*). Projections from the MGBv to primary auditory cortical areas are several orders of magnitudes more extensive than the projections from other thalamic nuclei, both in L1 and in the middle layers (*Figure 5C,D*). Closer inspection of the MGBv axons revealed that they tend to travel from the middle layers to L1 in columnar fashion, that is, in an almost straight line (*Figure 5C*,

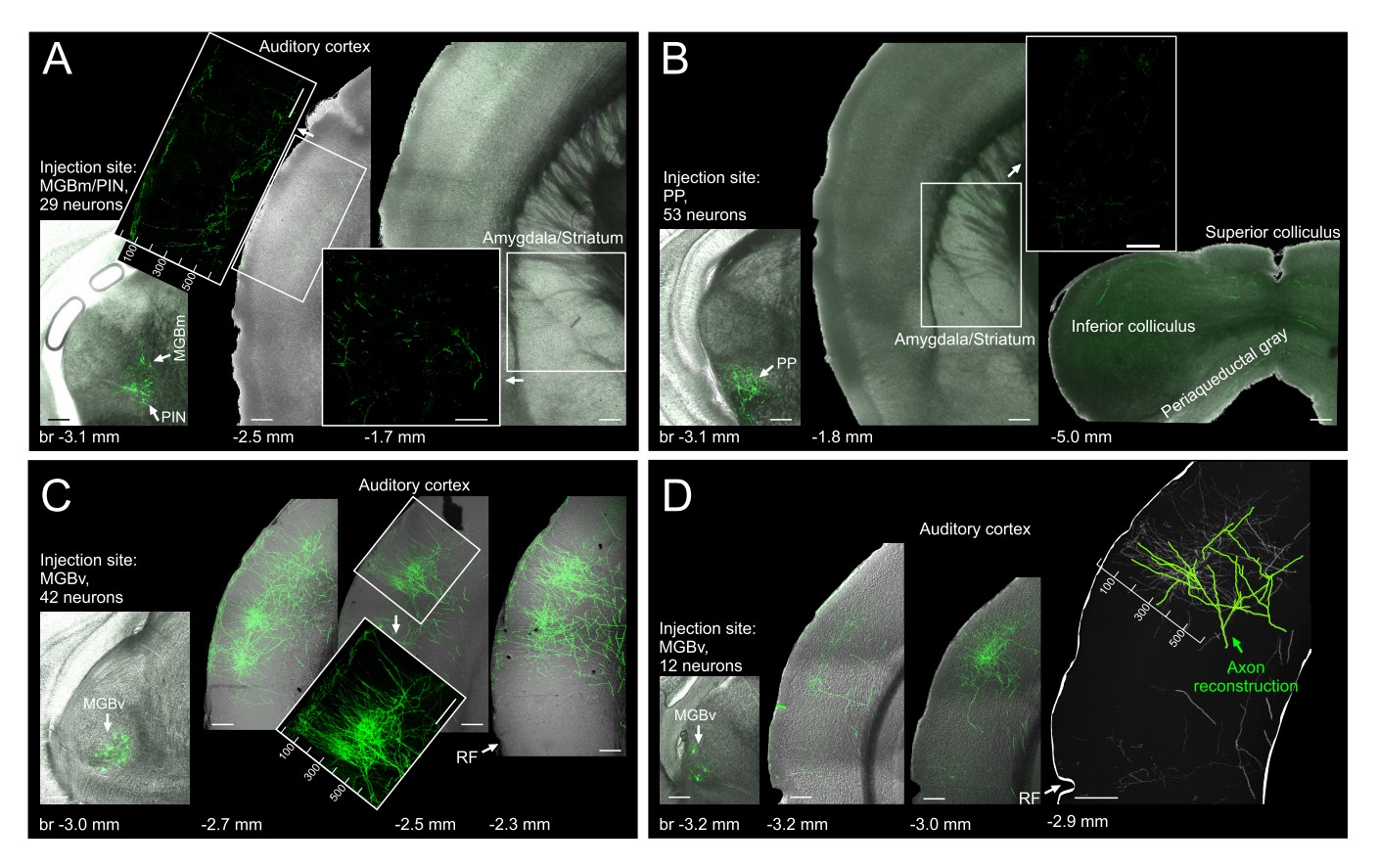

**Figure 5.** Of the different regions of auditory thalamus, only MGBv provides substantial input to primary auditory cortex. (**A**) Left, eGFP labelled neurons (green) in MGBm and PIN after very small injection of highly diluted AAV1.hSyn.Cre.WPRE.hGH and AAV1.CAG.Flex.eGFP.WPRE.bGH. Middle, thalamic axons in auditory cortex. Numbers indicate distance from cortical surface in μm. Right, thalamic axons in amygdala and striatum. (**B**) Left, eGFP labelled neurons following an injection in PP. Middle/Right, thalamic axons in amygdala, striatum and midbrain. No labelling was found in auditory cortex. (**C**) Left, eGFP labelled neurons following an injection in MGBv. Right, thalamic axons in auditory cortex. Numbers indicate distance from cortical surface in μm. Total rostrocaudal spread of thalamic axons in cortex exceeded 1 mm. (**D**) Left, eGFP labelled neurons in MGBv. Middle, thalamic axons in auditory cortex. Total rostrocaudal spread of thalamic axons in cortex exceeded 1 mm. Right, partial reconstruction of a single MGBv axon within a 100 μm thick section of auditory cortex. Numbers indicate distance from cortical surface in μm. Scale bars, 200 μm.

DOI: https://doi.org/10.7554/eLife.25141.009

inset), an arrangement which provides an anatomical substrate for our finding that L1 and L3b/4 thalamic input are in register tonotopically and exhibit a very similar organization. Furthermore, by partially reconstructing the axon from one MGBv neuron, we were able to confirm previous work in the rabbit (*Cetas et al., 1999*) showing how extraordinarily wide the arbors of MGBv axons tend to be, and that the same MGBv neurons provide input to different layers of the auditory cortex (*Figure 5D*).

Finally, we employed a cre-dependent virus in the CR-IRES-cre mice to express GCaMP6m exclusively in CR+ neurons outside the MGBv and performed two-photon calcium imaging of their axonal boutons in the auditory cortex of awake mice. These boutons typically responded very poorly to acoustic stimulation and, consequently, only very few FRAs were obtained that passed our inclusion criterion. In these experiments, we first identified A2 by its ventral location and the particularly dense thalamocortical axon labelling. Areas slightly dorsal of A2 were deemed to be in primary auditory areas. Five out of six imaged areas in primary auditory areas produced no FRAs at all, and one produced three FRAs. Even in A2, where the labelling was typically very dense (*Figure 4—figure supplement 1*), these numbers were very low. Here, the median number of FRAs obtained per imaged

area was 1 ± 3.5 (range: 0–9, n = 9 areas) suggesting that axon boutons with clearly defined FRAs are predominantly a feature of the projection from MGBv to the primary auditory cortical areas.

## Discussion

We have shown that while auditory cortical layers 1 and 3b/4 receive tonotopically matching thalamic input, the frequency selectivity of neighboring axon boutons is highly heterogeneous. That the thalamocortical projection is topographically arranged has been known for a long time (*Clark, 1936*; *Walker, 1937*). Until recently, however, it has not been possible to characterize the receptive fields of individual thalamic boutons (*Roth et al., 2016*), preventing any physiological assessment of how precisely organized this projection is.

We observed very similar patterns of results across two strains of C57BL/6 mice, one of which had the Cdh23[ahl] allele that otherwise predisposes this strain to age-related high frequency hearing loss corrected, and across both anesthetized and awake animals. The proportion of neurons with high BFs was greater in the C57BL/6NTac.*Cdh23*[753A>G] mice, potentially indicative of the beginning of high-frequency loss in the other animals. While the study was designed to capture the full extent of the thalamic input available to the auditory cortex, our own anatomical data together with the work of others (*Hackett et al., 2011*; *Llano and Sherman, 2008*), show that the organization we describe reflects almost exclusively the properties of the lemniscal thalamocortical projection from the MGBv to the primary cortical areas.

The presence of a well-defined tonotopic organization in the main thalamorecipient middle layers of auditory cortex (*Guo et al., 2012*; *Hackett et al., 2011*; *Winkowski and Kanold, 2013*) implies that the thalamic input should also be precisely arranged. Although the diffuse organization observed with axon bouton imaging contrasts with that expectation, it does provide an explanation for other findings. For instance, the observation that focal electrical stimulation of the MGB causes widespread activation of the cortex over several hundred micrometers (*Hackett et al., 2011*; *Kaur et al., 2005*) is easier to reconcile with a diffuse thalamacortical connectivity pattern in which similarly tuned thalamic axons, or even the same axon (*Cetas et al., 1999*), can connect with neurons located far apart in the auditory cortex. Similarly, the demonstration that thalamic inputs determine the bandwidth of the broadly tuned excitatory synaptic FRAs of auditory cortical neurons (*Liu et al., 2007*) can be explained more readily by our finding that most auditory cortical neurons have, within the boundaries of their dendritic trees (*Richardson et al., 2009*), access to thalamic terminals tuned to frequencies that collectively span several octaves.

Our results show that the cortical frequency map is built from a thalamic input map which is itself poorly organized. Thalamic projections synapse preferentially on spines within 100 µm of the soma of L3 and L4 neurons (*Richardson et al., 2009*), but how exactly these neurons integrate the available thalamic input to produce a more precisely ordered cortical frequency representation is unclear. Several mechanisms could contribute to this transformation. First, recent work in the visual cortex has shown that dendritic nonlinearities can affect the tuning of neurons (*Wilson et al., 2016*). Second, recurrent connections between cortical neurons, comprising over half of their inputs (*Lübke et al., 2000*), can amplify (*Happel et al., 2010*; *Li et al., 2013*) and may potentially homogenise (*Liu et al., 2007*) local tuning, especially if they are biased (*Cossell et al., 2015*). Finally, auditory cortical neurons may sample their thalamic inputs in a biased manner, similar to what has been proposed in the visual system (*Reid and Alonso, 1995*). However, given the broad synaptic tuning reported for thalamic inputs onto individual auditory cortical neurons — in rats the range of frequencies covered by the thalamic inputs onto a single L4 neuron lies between 3 and 5 octaves (*Liu et al., 2007*)— such biased connectivity seems less likely in the auditory thalamocortical system.

Although thalamic inputs primarily target the middle cortical layers, they innervate all cortical layers and particularly L1 (*Huang and Winer, 2000*; *Kimura et al., 2003*; *Kondo and Ohki, 2016*; *Roth et al., 2016*; *Rubio-Garrido et al., 2009*; *Sun et al., 2016*). Thalamic axons in L1 have different neuronal targets, mostly L1 inhibitory neurons (*Cruikshank et al., 2007*; *Ji et al., 2016*) and the apical dendrites of supra- and infragranular excitatory neurons (*Harris and Shepherd, 2015*; *Petreanu et al., 2009*), from those terminating in the middle layers, but whether the content of the information transmitted to different cortical layers also differs is not well understood. We found that L1 and L3b/4 inputs are fairly well matched tonotopically and show only minor differences in the degree of BF heterogeneity. This is consistent with two other studies which also found only minor

differences between the responses to oriented gratings of thalamic axons in L1 and L4 of visual cortex (**Kondo and Ohki, 2016**; **Sun et al., 2016**). Furthermore, our anatomical work revealed that many lemniscal thalamic axons travel from the middle layers up to L1 in a columnar fashion, a feature that helps explain why the properties of the thalamic input to L1 and to the middle layers are so similar. Traditionally, thalamic inputs to L1 and L3b/4 have been classified as belonging to separate channels, with L1 inputs described as matrix-type and L3b/4 input as core-type (**Clascá et al., 2012**; **Harris and Shepherd, 2015**; **Jones, 2001**). Yet, a number of single axon tracing studies in various species and cortical regions have described thalamic axons that form dense plexuses in L4 and project collaterals to L1 (**Cetas et al., 1999**; **Hashikawa et al., 1995**; **Kuramoto et al., 2009**; **Oda et al., 2004**). These and our current findings suggest that the laminar separation into matrix- and core-type inputs may not be so clear-cut. Nevertheless, L1 does receive a larger proportion of input from higher-order thalamic nuclei than L3b/4 (**Frost and Caviness, 1980**; **Linke, 1999**; **Linke and Schwegler, 2000**; **Llano and Sherman, 2008**; **Ryugo and Killackey, 1974**; **Smith et al., 2010**). Given that, in other sensory systems (**Roth et al., 2016**), input from higher-order thalamic nuclei has been shown to carry more motor and contextual sensory signals than the input from the first order nucleus, and that we observed generally poor responses to tone stimulation in higher-order thalamic axons, it is likely that recordings in behaving animals will reveal more pronounced differences between L1 and L3b/4 input.

A key question arising from our findings is why auditory thalamocortical projections are so imprecise. Precisely-organized tonotopic maps have been identified subcortically in the lemniscal part of the mouse inferior colliculus (**Barnstedt et al., 2015**; **Portfors et al., 2011**; **Stiebler and Ehret, 1985**), and anatomical and electrophysiological data indicate that the lemniscal thalamus is likely to be similarly organized (**Hackett et al., 2011**; **Lee and Sherman, 2010**; **Wenstrup, 2005**). Input from the dorsolateral geniculate nucleus to the visual cortex tends to be highly retinotopically ordered (**Roth et al., 2016**), so the mouse brain is capable of establishing and maintaining very precise connections between thalamus and cortex. This suggests that the diffuse topographic arrangement we observed in the auditory system may be functionally relevant. Broad spectral integration enables auditory cortical neurons to form representations of behaviorally-relevant sound sources (**Bar-Yosef et al., 2002**; **Las et al., 2005**). Furthermore, studies in different species have shown that auditory cortical frequency representations are highly plastic over multiple timescales (**Dahmen and King, 2007**), and individual neurons can rapidly change their stimulus selectivity with the behavioral context (**Fritz et al., 2003**). Such dynamic modulation of sound frequency processing can only be possible if cortical neurons have access to spectrally broad inputs (**Chen et al., 2011**; **Intskirveli et al., 2016**; **Metherate et al., 2005**; **Miller et al., 2001**; **Winer et al., 2005**). The organization of the thalamocortical projection revealed here is likely to be one part of the neural architecture underpinning this rapid plasticity and the cognitive flexibility it enables.

## Materials and methods

All experiments were approved by the local ethical review committee at the University of Oxford and licensed by the UK Home Office. Nine female C57BL/6 (Harlan Laboratories, UK) mice, five female C57BL/6NTac.$Cdh23^{753A>G}$ (MRC Harwell Institute, UK) mice and two female as well as one male B6(Cg)-$Calb2^{tm1(cre)Zjh}$/J ('CR-IRES-cre', Jackson Laboratories, CA, USA, Stock No: 010774) mice were used for calcium imaging. A further four female C57BL/6 (Envigo, UK) mice and one female B6(Cg)-$Calb2^{tm1(cre)Zjh}$/J were used for anatomical experiments.

### Virus transfection

As described before (**Barnstedt et al., 2015**) animals aged 4–6 weeks were premedicated with intraperitoneal injections of dexamethasone (Dexadreson, 4 μg), atropine (Atrocare, 1 μg) and carprofen (Rimadyl, 0.15 μg). General anesthesia was induced by an intraperitoneal injection of fentanyl (Sublimaze, 0.05 mg/kg), midazolam (Hypnovel, 5 mg/kg), and medetomidine (Domitor, 0.5 mg/kg). Mice were then placed in a stereotaxic frame (Model 900LS, David Kopf Instruments, CA, USA) equipped with mouth and ear bars, and located in a sterile procedure area. Depth of anesthesia was monitored by pinching the rear foot and by observation of the respiratory pattern. Body temperature was closely monitored throughout the procedure, and kept constant at 37°C by the use of a heating mat and a DC temperature controller in conjunction with a rectal temperature probe (FHC, ME, USA).

The skin over the injection site was shaved and an incision was made, after which a small hole of 0.5 mm diameter was drilled (Foredom K.1070, Blackstone Industries, CT, USA) into the skull with a 0.4 mm drill bit.

Viral injections were done using a pulled glass pipette and a custom-made pressure injection system. For calcium imaging experiments, C57BL/6 and C57BL/6NTac.$Cdh23^{753A>G}$ mice were injected with ~200 nl of AAV1.Syn.GCaMP6m.WPRE.SV40 (Penn Vector Core, PA, USA), diluted 1:2 in PBS, and B6(Cg)-$Calb2^{tm1(cre)Zjh}$/J mice were injected with ~200 nl of AAV1.Syn.Flex.GCaMP6m.WPRE. SV40 (Penn Vector Core), diluted 1:2 in PBS, into the right auditory thalamus. For anatomical experiments, one B6(Cg)-$Calb2^{tm1(cre)Zjh}$/J mouse was injected with ~200 nl of a 1:1 mixture of AAV1.Syn. GCaMP6m.WPRE.SV40 and AAV1.CAG.Flex.tdTomato.WPRE.bGH (Penn Vector Core). The stereotaxic coordinates were 2.9 mm posterior to bregma, 2.05 mm to the right of the midline and 3.0 mm from the cortical surface. Further anatomical experiments were carried out in C57BL/6 mice injected with very small amounts (<5 nl) of a 1:1 mixture of highly diluted (1:50000–100000 in PBS) AAV1. hSyn.Cre.WPRE.hGH (Penn Vector Core) and AAV1.CAG.Flex.eGFP.WPRE.bGH (Penn Vector Core). For these experiments, the stereotaxic coordinates were altered slightly from experiment to experiment in order to selectively target different subdivisions of the auditory thalamus.

The skin was then sutured and general anesthesia was reversed with an intraperitoneal injection of naloxone (1.2 mg/kg), flumazenil (Anexate, 0.5 mg/kg), and atipamezol (Antisedan, 2.5 mg/kg). Buprenorphine (Vetergesic, 1 ml/kg) and enrofloxacine (Baytril, 2 ml/kg) were injected postoperatively and again 24 hr later. In order to verify the successful transfection of neurons throughout the entire auditory thalamus, each mouse was killed at the end of the experiments and perfused transcardially, first with PBS and then with 4% paraformaldehyde in PBS. Mice used in anatomical experiments were euthanized and perfused three weeks after the virus injections. The relevant parts of the fixed brains were sectioned in the coronal plane at a thickness of 100 or 150 μm and images were taken with a Leica DMR upright fluorescence microscope or an Olympus FV1000 confocal microscope. Images were processed offline using ImageJ (NIH, MD, USA). Axonal reconstructions were carried out using a Leica DMR upright fluorescence microscope and Neurolucida (Microbrightfield, VT, USA) software.

## Window surgeries

For acute terminal imaging experiments, mice were premedicated with dexamethasone (4 mg/kg) and atropine (0.5 ml/kg), and general anesthesia was induced with ketamine (100 mg/kg, Vetalar) and medetomidine (140 μg/kg). The mouse was placed in a stereotaxic frame and body temperature was kept constant at 37°C. Both eyes were covered with eye ointment (Maxitrol, Alcon, TX, USA) to prevent corneal desiccation during anesthesia. A 2 cm flap of skin was cut to expose the parietal and temporal bones on the right hemisphere. The right temporalis muscle was separated from the temporal bone with a scalpel and partly removed. A 4.0 mm diameter region was marked on the right hemisphere, with its center ~2.5 mm posterior to bregma and ~4.5 mm to the right of the midline. Cyanoacrylate glue (Pattex Classic, Henkel, Germany) was applied to the surrounding skull, muscle, and wound margins to prevent further bleeding. A drill fitted with a 0.4 mm bit was used to thin the marked skull region and the central island of bone was removed to expose the underlying cortex. Saline was applied continuously for a few minutes to wash away any blood from the dura that could obscure imaging. Once all the bleeding stopped, a glass coverslip, 4.0 mm in diameter, was placed in direct contact with the surface of the cortex and attached to the edges of the skull with cyanoacrylate glue (Pattex Ultra Gel, Henkel). A small metal bar was attached to the skull over the left hemisphere with dental cement (Unifast Trad, GC Europe, Belgium), which was also used to cover all exposed areas of skull. The mouse was then placed on a custom-made stage, its head fixed to the stage using the steel bar.

To implant the cranial window and head bar in preparation for chronic, awake imaging experiments, anesthesia was induced with an intraperitoneal injection of fentanyl (Sublimaze, 0.05 mg/kg), midazolam (Hypnovel, 5 mg/kg) and medetomidine (Domitor, 0.5 mg/kg) and afterwards reversed with an intraperitoneal injection of naloxone (1.2 mg/kg), flumazenil (Anexate, 0.5 mg/kg), and atipamezol (Antisedan, 2.5 mg/kg). Buprenorphine (Vetergesic, 1 ml/kg) and enrofloxacine (Baytril, 2 ml/kg) were injected postoperatively and again 24 hr later. The head bar used for these experiments had a different shape, was larger, placed nearer the window and attached to the skull using Super-

Bond C&B (Sun Medical, Japan) dental acrylic. Mice were allowed to recover for at least one week before the first imaging session.

## Imaging

The imaging experiments were performed 3–6 weeks after making the virus injection. For anesthetized imaging, ketamine (50 mg/kg/h) and medetomidine (0.07 mg/kg/h) were regularly topped up at 30 min intervals to maintain a stable level of anesthesia throughout the experiment. For awake imaging, mice were placed inside a plexiglass body tube on a custom-made stage (*Guo et al., 2014*). All imaging took place inside a sound-attenuated chamber. A thin silicone tube coupled to an electrostatic loudspeaker (EC1, Tucker-Davis Technologies, FL, USA) was placed near the entrance of the mouse's left ear canal to deliver sounds during the experiment. The position of the tube was kept constant across imaging sessions. The drivers were calibrated using a G.R.A.S. 40DP (G.R.A.S., Denmark) microphone coupled to the tube to ensure a flat ($\pm$3 dB) response at all presented frequencies (1.25 to 80 kHz). Ambient noise was kept low by keeping the laser's power supply in a separate room. Sound generated by the resonant scanner was <40 dB SPL near the mouse's head. Stimuli were generated with an RZ6 processor (Tucker-Davis Technologies) and controlled through custom-written MATLAB (MathWorks, MA, USA) code.

To measure neuronal sound frequency sensitivity, we presented pure tones of 200 ms duration (with 5 ms raised cosine onset and offset ramps), which were varied randomly in frequency (from 1.25 to 80 kHz in 1/4 octave steps) and level (in 20 dB steps from 20 to 80 dB SPL based on measurements taken at the entrance to the ear canal in a mouse cadaver). They were presented at a rate of ~0.66 Hz (1 every 45 frames). This rate was similar to or slower than that used in previous, comparable, in vivo two-photon imaging studies (*Issa et al., 2014*; *Roth et al., 2016*; *Rothschild et al., 2010*), and was chosen because the calcium signal had usually fully decayed by the onset of the next stimulus. Using an even slower rate of ~0.5 Hz did not change the estimates of frequency tuning (data not shown). Each frequency-level combination was presented nine times. These 900 stimuli were presented in blocks of 300 allowing for the correction, between blocks, of any small drift in our imaging fields.

Imaging was performed using a commercial two-photon laser-scanning microscope (B-Scope, ThorLabs, VA, USA). Excitation light (930 nm) came from a SpectraPhysics Mai-Tai eHP (Spectra-Physics, CA, USA) laser fitted with a DeepSee prechirp unit (70 fs pulse width, 80 MHz repetition rate). The beam was directed into a Conoptics (CT, USA) modulator (laser power, as measured under the objective, varied from 10 to 50 mW) and scanned onto the brain with an 8 kHz resonant scanner (X) and a galvanometric scan mirror (Y). The resonant scanner was used in bidirectional mode, enabling the acquisition of 512 $\times$ 512 pixel frames at a rate of ~30 Hz. Emitted photons were guided through a 525/50 filter onto GaAsP photomultipliers (Hamamatsu, Japan). ScanImage (*Pologruto et al., 2003*) was used to control the microscope. Imaging was performed with a 40$\times$/0.80 NIR Apo immersion objective (Nikon, Japan). A motorised XYZ stage with a digital controller (ThorLabs) was used to record the coordinates of the imaged regions. Pictures of the vasculature were taken with a CCD camera (Lumenera, Canada) attached to the B-Scope and used, together with low-zoom two-photon images, for careful re-alignment of the window coordinates across imaging sessions. Reconstructed vasculature maps of the whole window were used for alignment of the electrophysiological recordings with the imaging sites.

## Electrophysiological recordings

After the final imaging session, we carried out extracellular electrophysiological cortical mapping experiments under anesthesia (ketamine 50 mg/kg/h + medetomidine 0.07 mg/kg/h) in each of the C57BL/6NTac.Cdh23$^{753A>G}$ mice to help with the identification of primary auditory cortical areas. After removal of the glass coverslip, 64 channel (8 $\times$ 8) probes (Neuronexus, MI, USA) were inserted to record from the middle layers of auditory cortex. Electrophysiological data were acquired on a RZ2 BioAmp processor (Tucker-Davis Technologies), and collected and saved using custom-written MATLAB (MathWorks) code (https://github.com/beniamino38/benware). Stimuli were generated using a RX6 Multifunction Processor (Tucker-Davis Technologies), amplified by a TDT SA1 Stereo Amplifier (Tucker-Davis Technologies), and delivered via a modified ultrasonic dynamic loudspeaker (Vifa, Avisoft Bioacoustics, Germany) coupled to a tube that was positioned near the entrance of the

mouse's left ear canal. They consisted of 200 ms pure tones spaced in one-third octave steps from 2 to 64 kHz at 40, 60 and 80 dB SPL.

## Data analysis

Data analysis was performed in MATLAB. Image stacks were registered to a 50-frame average using efficient subpixel registration methods (*Guizar-Sicairos et al., 2008*) to correct for *x–y* motion. Regions of interest (ROIs) were automatically extracted using a custom-written script implemented in MATLAB. Initially, each 512 × 512 pixel imaging area was parcellated into overlapping 8 × 8 pixel image patches. Next, a set of descriptors was calculated for each image patch. The descriptors used, 'Histograms of Oriented Gradients' (HOG; *Dalal and Triggs, 2005*), were extracted separately from each of the image patches and used as features for subsequent classification. After pre-training using manually annotated data, a support vector machine then used the HOG features of each image patch to determine whether it contained a bouton. The subset without boutons was discarded, whereas those classified as containing boutons were processed further. To draw the ROI masks for each image patch containing a bouton, a region-growing algorithm (*Nixon and Aguado, 2012*) was applied to each patch individually. The seed pixel for the region-growing algorithm was selected using a two-step procedure. First, a 'circular Hough transform' ('imfindcircles' MATLAB function) was applied to each image patch containing a bouton and a circle was drawn around the bouton. The brightest pixel within the circle was then used as a seed. After region growing, morphological erosion (*Nixon and Aguado, 2012*) was applied to each image patch, enhancing separation of overlapping bouton ROI masks. Finally, image patches were recombined into a single image containing all ROI masks. Once defined, all pixels within each ROI were averaged to give a single time course ($\Delta F/F$). This signal was high-pass filtered at a cutoff frequency of 0.03 Hz to remove slow fluctuations in fluorescence.

The first 15 frames (~500 ms) following stimulus onset were defined as the response window and a single-trial response was defined as the average $\Delta F/F$ within that window. ROIs were included for analysis only if they exhibited a statistically significant difference in response among the 100 frequency-level combinations (one-way ANOVA, p<0.001). For each ROI, a matrix of the averaged responses to different frequency-level combinations was constructed, with different levels arranged in rows and different frequencies arranged in columns. This matrix was then smoothed across frequencies using a three point wide running average. Best frequency (BF) was defined as the sound frequency associated with the highest response averaged across all sound levels. This measure of frequency preference is considered to produce the most orderly tonotopic maps (*Hackett et al., 2011*). In order to assess the tuning quality we fitted Gaussians to the level-averaged tuning curves (*Figure 1—figure supplement 1*). Co-tuning was defined as the standard deviation of a given BF distribution. The pairwise $\Delta BF$ was defined as the difference in BF in octaves between two boutons in the same imaged region. To determine whether the boutons' BFs varied along a particular axis within the brain (*Figures 2D* and *3F*), we correlated the BFs with their position on a series of axes spanning 360° at 1° intervals. The axis associated with the strongest positive correlation was taken as the direction of the tonotopic gradient. The tuning of multi-unit clusters was analyzed in a similar fashion to that of axonal boutons. The first 50 ms after stimulus onset were defined as the response window. Clusters were included for analysis only if they exhibited a statistically significant difference in response among the frequency-level combinations tested (one-way ANOVA, p<0.001). The BF was defined as the sound frequency associated with the highest spike count averaged across all sound levels.

## Statistics

Decisions on sample sizes were made on the basis of group sizes reported in published literature (e.g. *Roth et al., 2016*). Depending on the normality of distributions (Shapiro–Wilk test), parametric or non-parametric tests were used. All tests used are two-sided. Data are reported as median ±interquartile range unless stated otherwise. Effect size $r$ is defined as $r = \frac{z}{\sqrt{N}}$ (*Fritz et al., 2012*).

## Acknowledgements

This work was supported by a Clarendon scholarship from the University of Oxford to SAV-L, a Wellcome PhD studentship (WT102373/Z/13/Z) to YW, a Wellcome PhD studentship (WT105241/Z/14/Z) to ML and a Wellcome Principal Research Fellowship (WT07650AIA, WT108369/Z/2015/Z) to AJK. We thank the GENIE Program, Janelia Farm Research Campus and Howard Hughes Medical Institute for making GCaMP6 material available. We thank V Bajo for advice and we thank Michael Bowl at MRC Harwell for providing us with the C57BL/6NTac.$Cdh23^{753A>G}$ mice.

## Additional information

### Competing interests

Andrew J King: Senior Editor, eLife. The other authors declare that no competing interests exist.

### Funding

| Funder | Grant reference number | Author |
|---|---|---|
| University of Oxford | Clarendon Scholarship | Sebastian A Vasquez-Lopez |
| Wellcome | WT102372 | Yves Weissenberger |
| Wellcome | WT105241/Z/14/Z | Michael Lohse |
| Wellcome | WT07650AIA | Andrew J King |
| Wellcome | WT108369/Z/2015/Z | Andrew J King |

The funders had no role in study design, data collection and interpretation, or the decision to submit the work for publication.

### Author contributions

Sebastian A Vasquez-Lopez, Data curation, Software, Formal analysis, Validation, Investigation, Visualization, Methodology, Writing—original draft, Writing—review and editing; Yves Weissenberger, Data curation, Software, Formal analysis, Validation, Investigation, Visualization, Methodology, Writing—review and editing; Michael Lohse, Software, Formal analysis, Investigation, Visualization, Methodology, Writing—review and editing; Peter Keating, Resources, Software, Supervision, Methodology, Writing—review and editing; Andrew J King, Resources, Supervision, Funding acquisition, Writing—original draft, Project administration, Writing—review and editing; Johannes C Dahmen, Conceptualization, Data curation, Software, Formal analysis, Supervision, Validation, Investigation, Visualization, Methodology, Writing—original draft, Project administration, Writing—review and editing

### Author ORCIDs

Sebastian A Vasquez-Lopez http://orcid.org/0000-0003-4899-1221
Peter Keating http://orcid.org/0000-0002-0670-9075
Andrew J King http://orcid.org/0000-0001-5180-7179
Johannes C Dahmen http://orcid.org/0000-0001-9889-8303

### Decision letter and Author response

Decision letter https://doi.org/10.7554/eLife.25141.011
Author response https://doi.org/10.7554/eLife.25141.012

## Additional files

### Supplementary files

• Transparent reporting form
DOI: https://doi.org/10.7554/eLife.25141.010

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
