## [Decision Letter]

Thank you for submitting your article "Thalamic input to auditory cortex is locally heterogeneous but globally tonotopic" for consideration by *eLife*. Your article has been reviewed by two peer reviewers, and the evaluation has been overseen by a Reviewing Editor and Eve Marder as the Senior Editor. The following individuals involved in review of your submission have agreed to reveal their identity: Shihab Shamma (Reviewer #3).

The reviewers have discussed the reviews with one another and the Reviewing Editor has drafted this decision to help you prepare a revised submission.

The two reviewers highlight the major interest of your results. However, they point out that the results of several experiments require clarification (see reviewers' comments below). We therefore strongly encourage you to respond to the major concerns of the reviewers. They both agree on one particular experimental requirement, the labeling of lemniscal pathways, and suggest that you address this issue, in the revised form of your manuscript, in a mouse strain not affected by hearing loss at the time at which the central auditory pathway is studied. A mouse strain is now available, C57BL/6N genetically corrected for the auditory phenotype which should allow you to solve this problem (Genome Med. 2016 Feb 15;8(1):16. doi: 10.1186/s13073-016-0273-4).

Reviewer #2:

This interesting and important manuscript investigates the functional responses of thalamocortical axons in auditory cortex. The authors perform Ca^2+^ imaging from MGB terminals in auditory cortex and show that neighboring terminals can show very heterogeneous frequency tuning. They also describe the tonotopic organization in layer 1, which is new. Despite methodological shortcomings, these results are novel and this paper adds significantly to our understanding of auditory processing. The shortcomings center on the non-specificity of the experiments which make the interpretation of the experiments difficult. Multiple subnuclei of MGB are labelled and unspecified regions of auditory cortex are imaged.

Major points:

1) Methodological issues: a) The authors use C57BL/6 mice that at the time of the experiment are 9-12 weeks old. These mice are known to show hearing deficits at these ages which can alter the results.

b) The authors label both lemniscal and non-lemniscal nuclei in the MGB as well as areas outside the MGB (see Figure 1). Thus, the tuning heterogeneity observed could be due to intermingled non-lemniscal projections on a background of tonotopically organized lemniscal projections. This should be disambiguated or at least discussed in much more detail to highlight these serious limitations. The authors should also include photomicrographs of the injection sites of more animals. How consistent and how even is their labelling of the MGB? How did they determine the borders of the MGB in the fluorescence images?

c) The authors state that the image in "auditory cortex". The auditory cortex consists of many subfields but no detail is given about which cortical subfield the images were acquired. Given the functional diversity of auditory cortical fields (see Issa et al. 2014, 2017, Baba et al. 2016) the organization of thalamic terminals in the subfields might vary. To strengthen their case the authors should identify cortical subfields (e.g. by intrinsic or flavoprotein imaging) and then image terminals in A1 which would be expected to have the highest degree of tonotopic organization.

d) The authors only use female mice. This seems to be a curious choice given the general thought of higher variability in females due to the estrus cycle. Is the cortical organization different in males?

e) How many times are sound stimuli presented? How reliable are the responses? Are there differences between layer 1 and layer 4? Could varying reliability across terminals lead to uncertain BF determination?

2) The authors remark that inputs to layer 3b and 4 seem surprisingly similar. Given that layer 4 neurons can receive thalamic inputs on their soma but also at their proximal apical dendrites these labelled inputs could target the same layer 4 neurons. Thus, the similar organization of layers 3b and 4 does not seem to be surprising.

3) In general, the authors should elaborate more on the integrative properties of layer 4 neurons such as geometry and extent of their dendritic tree. These properties are crucial to transform the thalamic input map they see into a smooth functional map.

4) The Introduction posits that because one observes functional maps in the spiking output of layer 4 neurons the underlying thalamic input have to be organized.

5) Strawman since we know for decades that layer 4 integrates over thalamic input. For example in visual cortex graduated ocular dominance (e.g. binocular cells) emerge from monocular LGN inputs. Moreover, detailed in vivo path clamp recordings by the Zhang lab at USC have shown that layer 4 neurons receive broadly tuned thalamic input suggestive of integration over many differently tuned thalamic terminals. These issues should be flushed out more in the Introduction and Discussion (also see point above).

6) In the Introduction, the authors refer to anatomical evidence of precise tonotopic projections from MBG to A1. More detail about the spatial scale of these experiments (size and separation of injection sites) is required here to compare with the expectations and results of the current study.

Reviewer #3:

This is an excellently written report dealing with a very important question. The results are also striking in that they (apparently) contradict two previous cited (2-photon) studies that had shown an organized projection at least to the intermediate cortical layers. This work extends these results by looking directly at the tuning of the dendritic boutons in L1 and L3/4 which project from the thalamus. Clearly this study is technically superb despite its obvious limitations that the authors describe. So rather than going over the paper in detail, I will list below what I consider to be important questions that need to be clarified and discussed.

1) The two previous studies in mouse are consistent on the disorder of the upper layers, but were contradictory on the main tonotopic order in L3/4. Any issues with methodology that the authors can address to explain this disparity? Those studies reported that activity from the "neuropil" was tonotopically organized, which would argue that the axonal terminals are organized, and presumably also the tuning in the boutons.

2) As the authors discuss, the injections reflect all MGB nuclei, i.e., both Limniscal and non-limniscle (L and NL) pathways. I understand the present difficulties of labeling only the L pathway (which is tonotopically organized) so to what extent should this limitation affect the conclusions of the manuscript? That is, does it nullify the whole result? Can for example the L pathway be completely tonotopically organized, but that view be obscured by the influence of NL inputs? Is there a way to know the relative contributions of these 2 inputs so as to be certain that the results are at least indicative of the nontonotopicity of the L pathway?

3) One way to help disambiguate the two pathways might be to focus the analysis of the tuning correlations on the SHARPLY tuned boutons only. This assumes that the NL pathway is non-tuned and hence may help reveal a consistent (or clearer picture due to the L pathway alone). I think in all the analyses shown, simply labeling a bouton by one number (BF) without regard to how reliable this measurement is makes the analysis too noisy. After all, most of the time, the BF is unreliable to within 1/2 to 1 octave due to noise and broad tuning.

4) The measurements in Figure 1 reveal nicely isolated "spiking" or events. Why not exploit this to examine pairs of boutons that are highly spiking-correlated and hence presumably coming from the same axon? That would reveal if axons really spread their terminals across large distances or not. And then one can say something concrete about the topography? This would also apply to relating tonotopy to spatial spread.

5) Also one can ask and answer questions such as whether there is dependence in L1 versus L3/4 on either of the above that measures of tuning and inputs, and how they might correlate?

6) I did not get the sense that topography of the MGB projections to cortex can be (or is) addressed directly? For example, it really matters if the thalamic input is organized spatially in mouse? If it is organized, then one can explore the question of tonotopy independent of topography. For example, are we sure that the MGB in the mouse is strongly tonotopically organized? If not then perhaps the projections are organized but are not tonotopic because there is none in MGB. And so on.

7) Is this whole issue of a disordered mapping in cortex just a rodent issue? And is it possible that the tonotopy is "absent" in all areas up the auditory pathway in rodents?

---

## [Author Response]

Reviewer #2:This interesting and important manuscript investigates the functional responses of thalamocortical axons in auditory cortex. The authors perform Ca^2+^ imaging from MGB terminals in auditory cortex and show that neighboring terminals can show very heterogeneous frequency tuning. They also describe the tonotopic organization in layer 1, which is new. Despite methodological shortcomings, these results are novel and this paper adds significantly to our understanding of auditory processing. The shortcomings center on the non-specificity of the experiments which make the interpretation of the experiments difficult. Multiple subnuclei of MGB are labelled and unspecified regions of auditory cortex are imaged.Major points:1) Methodological issues: a) The authors use C57BL/6 mice that at the time of the experiment are 9-12 weeks old. These mice are known to show hearing deficits at these ages which can alter the results.

C57BL/6 mice are not normally considered to suffer from impaired high frequency hearing at the age used here (e.g. Ison et al. 2007), but there have been some reports that a decline in the number of neurons tuned to high frequencies can already be detected 1-2 months after birth, especially at higher levels of the auditory pathway such as the cortex (Willott et al., 1993). We therefore obtained the mouse strain recommended by the Reviewing Editor and carried out additional experiments. While we did not expect any fundamental differences in our results, we predicted that a larger proportion of high-frequency thalamic neurons in the C57BL/6NTac.*Cdh23^753A>G^* mice and, thus, a broader overall BF distribution would be reflected in even weaker co-tuning and, thus, higher frequency heterogeneity than in the young adult C57BL/6 mice we used previously. That is exactly what we found: Slightly broader overall BF distribution and even weaker co-tuning in individual imaged areas. And just as before, we found that thalamocortical inputs to cortical layer (L)1 and L3b/4 are in register tonotopically. Furthermore, in order to address another concern voiced by the reviewer, we carried out these additional experiments in awake rather than anesthetized animals and followed up these experiments with an electrophysiological cortical frequency mapping experiment to help us attribute individual imaging regions to particular cortical areas.

To describe these new results, we added a figure (new Figure 3) and the following paragraphs to the Results section:

“The C57BL/6 strain employed in the above experiments is the most popular laboratory mouse strain, and is used as genetic background for the overwhelming majority of genetically modified mouse strains, the availability of which make this species such a useful model system for neuroscience research. […] These recordings demonstrated that the vast majority (18/20) of imaging regions were located in the primary cortical areas.”

b) The authors label both lemniscal and non-lemniscal nuclei in the MGB as well as areas outside the MGB (see Figure 1). Thus, the tuning heterogeneity observed could be due to intermingled non-lemniscal projections on a background of tonotopically organized lemniscal projections. This should be disambiguated or at least discussed in much more detail to highlight these serious limitations. The authors should also include photomicrographs of the injection sites of more animals. How consistent and how even is their labelling of the MGB? How did they determine the borders of the MGB in the fluorescence images?

This is an important point and we agree that it needs to be addressed properly. We therefore carried out a number of new experiments to better understand the contributions of lemniscal, non-lemniscal and paralaminar auditory thalamic nuclei to the thalamocortical projection.

First, we showed that the CR-IRES-cre mouse line is suitable for targeting exclusively neurons outside the lemniscal auditory thalamus, i.e. outside MGBv. Then we show that input to cortex from these neurons is restricted mostly to secondary auditory cortex (particularly area A2), that the few axons found in primary auditory cortical areas are restricted mostly to layer 1, and that the even fewer axons found in the middle layers are located primarily below the main thalamic input, i.e. in upper layer 5.

Second, we carried out a large number of experiments involving the transfection of very small numbers of neurons through minute (<5 nl) injections of a mixture of highly diluted (1:50000-100000 in PBS) cre-expressing AAV and cre-dependent eGFP-expressing AAV in different parts of the auditory thalamus. We have included the results from a subset of these experiments in which we managed to label a small number of neurons in a defined subsection of the auditory thalamus. These experiments confirm and extend the above results by showing that: a) Projections from the medial part of the auditory thalamus (MGBm/PIN) provide only extremely sparse input to auditory cortex, which primarily terminates in L1, and otherwise is located below the layer(s) in which input from the MGBv is densest. b) Projections from the peripeduncular nucleus (PP) do not enter the auditory cortex and instead remain subcortical. c) Projections from the MGBv to the primary auditory areas are several orders of magnitudes more extensive than the projections from other thalamic nuclei, both in L1 and in the middle layers. d) Many MGBv axons travel from the middle layers to L1 in an almost straight line, an arrangement that provides an anatomical substrate for our finding that L1 and L3b/4 thalamic input are in register tonotopically and exhibit a very similar organisation. By partially reconstructing the axon from one MGBv neuron, we confirm previous work in the rabbit (Cetas et al., 1999) showing how extraordinarily wide the arbours of MGBv axons tend to be and that the same MGBv neurons provide input to different layers of the auditory cortex.

We now show a total of 8 photomicrographs of 5 animals. The labelling was good throughout the thalamus, although there is some variability in the brightness of the fluorescence which correlates with cell density as is evident from the DAPI images in Figure 4.

Finally, by employing the CR-IRES-cre line to perform functional imaging of thalamic boutons originating exclusively from outside the MGBv, we found that only a very small minority of these boutons respond to the stimuli we used and very few FRAs were obtained that pass our inclusion criterion: Less than one on average per imaging region in primary auditory areas.

To summarise, our new anatomical data, together with other work (e.g. Llano and Sherman, 2008; Hackett et al., 2011), show that the non-lemniscal and paralaminar auditory thalamic nuclei provide either no or very sparse input to primary auditory cortical areas, and particularly to the middle cortical layers. Furthermore, the functional imaging of boutons on axons originating from outside the MGBv revealed that very few of these boutons respond to the stimuli we employed. Consequently, we can conclude that the organization we describe reflects almost exclusively the properties of the lemniscal thalamocortical projection, particularly in those cases where we can demonstrate that the imaging was carried out in primary cortical areas.

We added two new figures (Figure 4 and Figure 5) and the following text to the manuscript:

“The auditory thalamus consists of several subnuclei. Besides the ventral division of the MGB (MGBv), which is the largest subnucleus and part of the lemniscal pathway, these are the non-lemniscal dorsal division of the MGB (MGBd) and the paralaminar nuclei – the medial division of the MGB (MGBm), the posterior intralaminar nucleus (PIN), the suprageniculate nucleus (SG) and peripeduncular nucleus (PP). […] Even in A2, where the labelling was typically very dense (Figure 4—figure supplement 1), these numbers were very low. Here, the median number of FRAs obtained per imaged area was 1 + 3.5 (range: 0-9, n = 9 areas) suggesting that axon boutons with clearly defined FRAs are predominantly a feature of the projection from MGBv to the primary auditory cortical areas.”

c) The authors state that the image in "auditory cortex". The auditory cortex consists of many subfields but no detail is given about which cortical subfield the images were acquired. Given the functional diversity of auditory cortical fields (see Issa et al. 2014, 2017, Baba et al. 2016) the organization of thalamic terminals in the subfields might vary. To strengthen their case the authors should identify cortical subfields (e.g. by intrinsic or flavoprotein imaging) and then image terminals in A1 which would be expected to have the highest degree of tonotopic organization.

We already reported one case in the original manuscript in which we can show, based on the tonotopic gradient, that the imaging was done in A1. We have added further data from terminals imaged in the primary areas (A1 and AAF). To help with cases where it wasn’t clear from the imaging data alone which field we were in, we followed up the imaging with an electrophysiological mapping experiment.

There are several reasons why we are confident that the overwhelming majority of our data (including the original dataset, where we were not always able to verify the location of the imaging field) were obtained from primary auditory areas. First, we imaged only where we could see robust labelling in L3b/4. Given that thalamic input is far denser in the primary areas than outside of those areas (as shown, for example, in the new Figure 4), the location of our imaging fields should have been strongly biased to those primary areas. Second, our imaging windows tended to be too dorsal to include much of A2, which is the largest of the mouse’s secondary auditory areas and lies at the ventral tip of auditory cortex (for imaging of the CR-IRES-cre line, we deliberately placed the windows slightly more ventral in order to target A2). Third, the imaging fields identified to be within A1 (based on the direction of the tonotopic gradient across the cortex) show co-tuning very similar to the rest of the dataset.

d) The authors only use female mice. This seems to be a curious choice given the general thought of higher variability in females due to the estrus cycle. Is the cortical organization different in males?

Given the smaller variability in body/brain size among female mice, stereotaxic injections are somewhat easier to perform in females than in males. Furthermore, other studies, including those exploring cortical tonotopy, have often used exclusively female mice (Hackett et al., 2011; Guo et al., 2012; Joachimsthaler et al., 2014), so we do not think that this is a curious choice. Finally, we have used both male and female mice in many other projects involving calcium imaging, electrophysiology and behavioural tests and have not come across any obvious differences.

e) How many times are sound stimuli presented? How reliable are the responses? Are there differences between layer 1 and layer 4? Could varying reliability across terminals lead to uncertain BF determination?

As stated in the Materials and methods section, each stimulus was presented 9 times. This is comparable to what other calcium imaging studies have used (Bandyopadhyay et al., 2010: 6-15 repeats; Rothschild et al., 2010: 8 repeats; Winkowski and Kanold, 2013: 10 repeats; Issa et al., 2014: 3-10 repeats). The frequency spacing (0.25 octaves) was identical to that used by most of the above studies. A further step we took to ensure a robust estimation of BFs was to present stimuli at several (4) different sound levels, something which was only done by two of the aforementioned studies, but has been shown to produce much more robust tonotopic maps (Guo et al., 2012). Moreover, we used a conservative inclusion criterion (p < 0.001) to avoid including unreliably responding boutons in the dataset. Finally, we confirmed that the level-averaged tuning curves from which we determined the BFs could generally be well approximated by a Gaussian function.

In order to measure response reliability, we took the same approach as in one of our previous studies (Barnstedt et al., 2015), in which we investigated the response properties of inferior colliculus neurons and corticocollicular axon boutons. Specifically, we measured the similarity between a bouton’s responses to each of the nine presentations of its best stimulus (the frequency-level combination that evoked the strongest average response) by calculating the average correlation coefficient for all pairwise combinations of the nine responses. Here, ‘response’ is defined as the entire 45-frame trace snippet post stimulus onset. As expected, the average reliability coefficient we measured for thalamic boutons (~0.15) was higher than that measured for cortical boutons (~0.08), but lower than that in the inferior colliculus (~0.34).

There was a small difference in response reliability between L1 and L3b/4, with L1 boutons being marginally but statistically significantly more reliable than L3b/4 boutons (0.155 vs. 0.138). If this difference were to lead to more uncertain BF estimates in L3b/4 than L1, we would expect to see more heterogeneous BF distributions in L3b/4 than in L1. However, this was not the case.

In summary, there is nothing to suggest that any aspect of our stimulation paradigm or the signals we measured may have caused an unusual degree of uncertainty in BF determination. These details are all in the Materials and methods section.

2) The authors remark that inputs to layer 3b and 4 seem surprisingly similar. Given that layer 4 neurons can receive thalamic inputs on their soma but also at their proximal apical dendrites these labelled inputs could target the same layer 4 neurons. Thus, the similar organization of layers 3b and 4 does not seem to be surprising.

We agree. We did not expect to see any difference between layer 3b and 4 inputs, but thought it would be informative to include this finding.

3) In general, the authors should elaborate more on the integrative properties of layer 4 neurons such as geometry and extent of their dendritic tree. These properties are crucial to transform the thalamic input map they see into a smooth functional map.4) The Introduction posits that because one observes functional maps in the spiking output of layer 4 neurons the underlying thalamic input have to be organized.5) Strawman since we know for decades that layer 4 integrates over thalamic input. For example in visual cortex graduated ocular dominance (e.g. binocular cells) emerge from monocular LGN inputs. Moreover, detailed in vivo path clamp recordings by the Zhang lab at USC have shown that layer 4 neurons receive broadly tuned thalamic input suggestive of integration over many differently tuned thalamic terminals. These issues should be flushed out more in the Introduction and Discussion (also see point above).

Points 3-5 are closely related so we address them together.

We already cite many of the studies relevant to the question how auditory cortical neurons integrate thalamic input (e.g. Richardson et al., 2009, as well as Liu et al., 2007 and Li et al., 2013 from the Zhang/Tao lab) and included a paragraph in the Discussion devoted to this topic. More specifically, we already made the very point about thalamocortical integration that the reviewer refers to: “given the broad synaptic tuning reported for thalamic inputs onto individual auditory cortical neurons – in rats the range of frequencies covered by the thalamic inputs onto a single L4 neuron lies between 3 and 5 octaves (Liu et al., 2007)…”).

Nevertheless, we have added an extra sentence to the Introduction to deal with this point:

“Indeed the frequency tuning of thalamic inputs that converge onto individual auditory cortical neurons can span several octaves (Liu et al., 2007), suggesting a need for integration across differently tuned afferent terminals.”

We have also added the following:

“Thalamic projections synapse preferentially on spines within 100 µm of the soma of L3 and L4 neurons (Richardson et al., 2009) but how exactly these neurons integrate the available thalamic input to produce…”

6) In the Introduction, the authors refer to anatomical evidence of precise tonotopic projections from MBG to A1. More detail about the spatial scale of these experiments (size and separation of injection sites) is required here to compare with the expectations and results of the current study.

The point we wished to make in the Introduction [“Although retrograde tracing of thalamocortical inputs (Brandner and Redies, 1990; Hackett et al., 2011) suggests strict topography, anterograde tracing (Huang and Winer, 2000) and reconstruction of single thalamic axons (Cetas et al., 1999) indicate considerable divergence in the auditory thalamocortical pathway.”] is that the anatomical evidence is conflicting, i.e. that retrograde tracing suggests precise topography whereas anterograde tracing implies the opposite. This is not a novel observation (see review by Winer et al., 2005, cited in the manuscript) and we don’t think the manuscript would benefit from including the methodological information requested by the reviewer. An interested reader will be able to find the information in the cited papers. In fact, the two retrograde tracing studies that we cited were quite different in their methodology, e.g. in the species and tracers used and the quantities injected (Brandner and Redies worked on cats and used several different retrograde tracers, some of which were ‘injected’ in crystalline form (HRP, Bisbenzimid) and others (nuclear yellow) in liquid form (0.05 µl to 0.1 µl), whereas Hackett et al. worked in mice, used CTB and injected 0.1-0.2 µl). Nevertheless, they came to very similar conclusions, suggesting that the precise methodological details are not particularly relevant. What matters is that anterograde tracing studies (and the in vivo patch clamp recordings referred to by the reviewer in the previous point) suggest very different conclusions about the organization of the thalamcortical projection, highlighting the need for the present study using in vivo calcium imaging of thalamocortical axon terminals.

Reviewer #3:This is an excellently written report dealing with a very important question. The results are also striking in that they (apparently) contradict two previous cited (2-photon) studies that had shown an organized projection at least to the intermediate cortical layers. This work extends these results by looking directly at the tuning of the dendritic boutons in L1 and L3/4 which project from the thalamus. Clearly this study is technically superb despite its obvious limitations that the authors describe. So rather than going over the paper in detail, I will list below what I consider to be important questions that need to be clarified and discussed.1) The two previous studies in mouse are consistent on the disorder of the upper layers, but were contradictory on the main tonotopic order in L3/4. Any issues with methodology that the authors can address to explain this disparity? Those studies reported that activity from the "neuropil" was tonotopically organized, which would argue that the axonal terminals are organized, and presumably also the tuning in the boutons.

In the manuscript we refer to four studies (Bandyopadhyay et al., 2010; Issa et al., 2014; Rothschild et al., 2010; Winkowski and Kanold, 2013) that investigated the tonotopic organisation of the mouse auditory cortex with two-photon calcium imaging. Three, including the one that shows the strongest evidence for tonotopic order in L4 (Winkowski and Kanold, 2013), used synthetic indicators that are injected into the cortex and could have been taken up by axons at the injection site including thalamocortical ones. However, the thalamic axons should be far outnumbered by axons and dendrites belonging to cortical neurons even in L3b/4, so the contribution made by thalamic axons to the neuropil signal is likely to be very minor in these studies. The fourth study (Issa et al., 2014), which found a particularly high degree of tonotopic order, used transgenic mice expressing calcium indicators in particular cell types. One of the transgenics used expressed GCaMP3 in EMX1 positive neurons. In these mice, no or very few thalamic neurons express GCaMP and so their axons would not have contributed to the neuropil signal picked up in the cortex. Nevertheless, different amounts of neuropil signal and different amounts of ‘contamination’ of the somatic signals by the neuropil signal might explain differences between some studies. However, thalamic axons are not likely to play a significant role.

2) As the authors discuss, the injections reflect all MGB nuclei, i.e., both Limniscal and non-limniscle (L and NL) pathways. I understand the present difficulties of labeling only the L pathway (which is tonotopically organized) so to what extent should this limitation affect the conclusions of the manuscript? That is, does it nullify the whole result? Can for example the L pathway be completely tonotopically organized, but that view be obscured by the influence of NL inputs? Is there a way to know the relative contributions of these 2 inputs so as to be certain that the results are at least indicative of the nontonotopicity of the L pathway?3) One way to help disambiguate the two pathways might be to focus the analysis of the tuning correlations on the SHARPLY tuned boutons only. This assumes that the NL pathway is non-tuned and hence may help reveal a consistent (or clearer picture due to the L pathway alone). I think in all the analyses shown, simply labeling a bouton by one number (BF) without regard to how reliable this measurement is makes the analysis too noisy. After all, most of the time, the BF is unreliable to within 1/2 to 1 octave due to noise and broad tuning.4) The measurements in Figure 1 reveal nicely isolated "spiking" or events. Why not exploit this to examine pairs of boutons that are highly spiking-correlated and hence presumably coming from the same axon? That would reveal if axons really spread their terminals across large distances or not. And then one can say something concrete about the topography? This would also apply to relating tonotopy to spatial spread.5) Also one can ask and answer questions such as whether there is dependence in L1 versus L3/4 on either of the above that measures of tuning and inputs, and how they might correlate?

Points 2-5 all concern the question to which extent the organization described in our manuscript reflects that of the lemniscal thalamocortical projection or additional contributions from the input originating in other thalamic nuclei and are therefore addressed together. We appreciate the suggestion to use tuning properties to disambiguate between inputs from different thalamic sources. However, the existing evidence suggests that the tuning properties of neurons from different thalamic subnuclei (Anderson and Linden, 2011) can be very similar, making this approach unlikely to produce clear-cut results. Furthermore, we acknowledge the suggestion to use correlations between boutons to identify pairs belonging to the same axon and the distances between them. However, the problem with this approach is that two boutons may be highly correlated without belonging to the same axon and, even more importantly, that this approach would be limited to the size of the individual imaging regions, i.e. about 100 µm.

In order to properly address the extremely important question of to what extent the functional organization we observe reflects the properties of the lemniscal pathway or projections from other thalamic subnuclei, we have devoted a lot of time and effort to gathering additional anatomical and functional data. These data, which are shown in the new Figure 4 and Figure 5 have largely resolved this question by allowing us to demonstrate the relative contributions of lemniscal, non-lemniscal and paralaminar auditory thalamic nuclei to the thalamocortical projection. As a result, we believe our manuscript has been greatly strengthened (The point was also raised by reviewer #2 so please see our response given to reviewer #2 point 1b.)

6) I did not get the sense that topography of the MGB projections to cortex can be (or is) addressed directly? For example, it really matters if the thalamic input is organized spatially in mouse? If it is organized, then one can explore the question of tonotopy independent of topography. For example, are we sure that the MGB in the mouse is strongly tonotopically organized? If not then perhaps the projections are organized but are not tonotopic because there is none in MGB. And so on.

While the anatomical and electrophysiological evidence certainly suggests that the mouse MGB is tonotopically organized (e.g. Horie et al., 2013; Hackett et al., 2011), we are not aware of any imaging data that could speak to the level of tonotopy in the mouse MGB at the fine sale. The only published two-photon imaging study of a subcortical auditory structure is one of our own studies (Barnstedt et al., 2015). Based on the strict tonotopic organization that we found to be present in the central nucleus of the mouse inferior colliculus (IC), it seems highly likely that the next stage of the lemniscal pathway, the MGBv, is also highly tonotopically organized even if the level of organisation may not be as strict as that in the IC. To answer the question of whether the thalamocortical projections are organized, we can look at published as well as our own new anatomical data presented in the revised manuscript. Cetas et al. (1999), for instance, reconstructed individual axons projecting from the MGBv to the auditory cortex in rabbits and showed that their arbors tend to be extraordinarily wide and project to parts of the cortex that are millimetres apart. Our own anatomical work suggests that the projection patterns of mouse MGBv axons, taking account of the smaller brain size, are very similar. So to summarise, the existing evidence favours an explanation in which the coarse tonotopic organization evident in the thalamic input is caused by the projection patterns of thalamocortical axons rather than a poorly organised MGBv.

In the revised manuscript we added the following sentence:

“Precisely-organized tonotopic maps have been identified subcortically in the lemniscal part of the mouse inferior colliculus (Barnstedt et al., 2015; Portfors et al., 2011; Stiebler and Ehret, 1985), and anatomical and electrophysiological data indicate that the lemniscal thalamus is likely to be similarly organized (Hackett et al., 2011; Lee and Sherman, 2010; Wenstrup, 2005).”

7) Is this whole issue of a disordered mapping in cortex just a rodent issue? And is it possible that the tonotopy is "absent" in all areas up the auditory pathway in rodents?

Our own two-photon calcium imaging experiments in ferret auditory cortex (Gaucher, Panniello, Dahmen, King, Walker 2017, ARO MWM abstract PS460) suggest that the fine-scale tonotopic organisation in rodents and carnivores is similar at the cortical level.

In Barnstedt et al., 2015 we performed two-photon calcium imaging in the central nucleus of the inferior colliculus of the mouse and found that the neurons there were very tightly tonotopically ordered, so strict tonotopy is certainly present up to at least the level of the inferior colliculus. This point is addressed for the thalamus in the previous comment. We therefore don’t think that this is a mouse issue. In proposing at the end of the Discussion that the imprecise organization of the thalamocortical projection could subserve spectral integration and task- or use-dependent plasticity in the cortex, we cited studies from different species (including cats and the reviewer’s own work in ferrets). To emphasize that point, we have added ‘in different species’ to the following sentence:

“Furthermore, studies in different species have shown that auditory cortical frequency representations are highly plastic over multiple timescales (Dahmen and King, 2007), and individual neurons can rapidly change their stimulus selectivity with the behavioral context (Fritz et al., 2003).”